# Characterization and Analysis of Corrosion Resistance of Rubber Materials for Downhole Tools in a High-Stress Environment with Coupled H_2_S-CO_2_

**DOI:** 10.3390/ma17040863

**Published:** 2024-02-12

**Authors:** Leilei Gong, Yulin Chen, Meng Cai, Junliang Li, Qiuyu Lu, Duo Hou

**Affiliations:** 1School of Mechanical Science and Engineering, Northeast Petroleum University, Daqing 163000, China; 2Research Institute of Oil Production Engineering, Daqing Oilfield Co., Daqing 163000, China; cyycaimeng@petrochina.com.cn (M.C.); bullsliang1982@163.com (J.L.); luqiuyu@petrochina.com.cn (Q.L.); 3Heilongjiang Oil and Gas Reservoir Production and Increase Focus Laboratory, Daqing 163000, China; 4The First Oil Production Plant of Daqing Oilfield Co., Ltd., Daqing 163000, China; yulinchen_a@petrochina.com.cn; 5School of Oil and Gas Engineering, Southwest Petroleum University, Chengdu 610000, China; dragon-duo@hotmail.com

**Keywords:** acid pressing condition, downhole tools, rubber material, corrosion resistance, temperature, sealing performance

## Abstract

In the process of constructing deep natural gas wells in Sichuan and Chongqing, gas wells encounter various technical challenges such as high temperature, high pressure, and a corrosive environment containing H_2_S and CO_2_. The corrosion of rubber materials in these acidic environments can easily lead to seal failure in downhole tools. To better investigate the corrosion resistance of rubber materials in acidic environments, we utilized a dynamic cyclic corrosion experimental device capable of simulating the service conditions experienced by downhole tools under high-temperature, high-pressure multiphase flow. Corrosion-resistance tests were conducted on fluororubbers (FKM) 1, 2, 3, and HNBR (hydrogenated nitrile-butadiene rubber) under acidic conditions (80 °C and 160 °C), along with sealing corrosion tests on O-rings. These tests aimed to analyze the mechanical properties, hardness, and corrosion resistance before and after exposure to acid media as well as the sealing performance of O-rings. Ultimately, our goal was to identify suitable rubber materials for acidic pressure environments. Experimental results revealed that all four types of rubber exhibited decreased elongation at break after undergoing corrosion testing; however, fluororubber 3 demonstrated significant susceptibility to temperature effects while the other three types showed minimal impact from temperature variations. Fluororubber 1 and fluororubber 3 displayed substantial deformation levels whereas mechanical properties greatly deteriorated for fluororubber 2. Overall, HNBR showcased superior comprehensive performance.

## 1. Introduction

Natural rubber (NR) is a well-known biopolymer. Polybutadiene (PB) and poly(styrene-butadiene-styrene) (SBS) are industrial rubbers with a diverse field of applications, such as in the soles of shoes, sealing rings, gaskets, damping materials, insulation materials, antivibration bushes, automotive parts, and the tire industry, among others, where elastomeric properties and durability are essential [1,2,3]. SBS, SBR, and PB are mostly used to manufacture of tires. Nitrile–butadiene rubber (NBR), hydrogenated nitrile-butadiene rubber (HNBR), fluoroelastomer (FKM), and Tetrapropyl fluororubber (FEPM) have good mechanical properties, and can be used in room temperature and high temperature environment sealing [4,5,6].

With the continuous development of highly acidic gas fields, the Wellbore service environment faces challenges such as high temperature, high pressure, and high acidity [7]. The downhole tool, the rubber barrel, plays a crucial role in generating contact pressure with the pipe string to prevent interaction between formation fluid, interlayer fluid, and pressure. However, corrosion in highly corrosive media can decrease the physical and mechanical properties of the rubber cylinder, leading to swelling and aging of the material. This results in loss of its high elastic sealing ability and weakening of contact stress between it and the pipe string. Ultimately, this leads to increased pressure in the annulus which poses serious threats to safe production of oil and gas wells while also increasing exploitation costs [8].

In recent years, Zeng D et al. [9] have demonstrated that tetrapropyl fluororubbers (AFLAS) rubber exhibit significant hardening in both the gas and liquid phases, along with weakened tensile properties, reduced hardness, and increased permanent deformation under high-acid gas well conditions (pressure: 60 MPa, temperature: 175 °C, H_2_S concentration: 20 vol%, CO_2_ concentration: 5 vol%). Research findings by Liu Jianxin et al. [10] revealed that rubber expands under high temperature and high-pressure oil/water/CO_2_ environments, leading to damage in the cross-linked structure, increased mass and volume, as well as decreased mechanical properties such as hardness, tensile strength, and elongation upon breaking. HNBR rubber H1010 exhibited better CO_2_ corrosion resistance compared to tetrafluororubber 100S under test conditions, making it suitable for CO_2_ flooding development environments in oilfields. Wang Jinchang et al. [11] investigated the corrosion damage of tetrafluororubber O-rings under wellbore conditions. After exposure to a corrosive environment with pressure, the strength of tetrafluororubber O-rings significantly decreased leading to brittle fractures. Zhang Rui et al. [12] research results indicated that rubber is prone to aging when exposed to environments with high CO_2_ content. Additionally, higher temperatures intensify the corrosive effects of media and accelerate rubber aging rates. Zhu Dajiang et al.’s research found that the crosslinking and degradation of rubber molecular chains due to aging in a high temperature and high-pressure environment lead to the deterioration of mechanical properties, thereby affecting the sealing performance of the rubber compound [13]. Rapid decompression results in gas dissolved in the rubber material not overflowing in time, leading to swelling, deformation, bubbles formation and even cracking of the rubber material [14]. FKM, fluorosilicone rubber (FVMQ), FEPM and HNBR exhibit decreased mechanical properties (reduced tensile strength and elongation at break; decreased hardness; increased compression permanent deformation) in corrosive environments. Among them, HNBR shows the lowest decrease compared with its initial state indicating excellent elastic properties and good stability. HNBR contains a relatively stable cross-linked network structure that maintains good mechanical properties when exposed to acidic corrosive media under high temperature and pressure conditions [15,16]. 

Aleksandra Smejda-Krzewicka et al. applied HNBR rubber to synthesize new materials, which improved the applicability of rubber materials [17]. Shaw, Barnabas et al. [18] Analysis of the DSC, FTIR and TGA results confirm that during ageing there is a steady evolution of the some of the ingredients in the compound such as the processing oils and the plasticizer which appears to have a significant effect on the fatigue properties of the HNBR materials.According to the results obtained, the tensile strength and hardness of the obtained HNBR/CR/Ag_2_O vulcanizate increased proportionally with the increase of CR content, while the tearing strength showed an inverse relationship. The resulting novel unconventional materials have significant thermal oxidation resistance, which is confirmed by the high aging coefficient. Ziwen Cui et al. [19], focused on the effect of the amount of crosslinking agent on the morphology and properties of HNBR/TPEE-TPVs. With the increase of peroxide dosage, the crosslinking degree of HNBR was increased, the HNBR domain size was smaller, the distribution was more uniform, and the rubber network was stronger. As a result, the elasticity and high temperature oil resistance of TPV increase, but its molten state fluidity, tensile properties and heat resistance decrease. The investigation demonstrates that as the DV time increases from 2 to 6 min, the rubber network progressively reinforces and the HNBR domain size decreases under more adequate shearing action, resulting in a more homogeneous morphology [20].

Rubber is in different states at different temperatures, so the rubber material exhibits different mechanical properties. In order to effectively ensure the service conditions of rubber materials under high temperature and high-pressure conditions containing H_2_S and CO_2_, this study employs a high-temperature autoclave to simulate the corrosive environment with H_2_S and CO_2_ in different well sections (at temperatures of 80 °C and 160 °C). Subsequently, corrosion resistance and sealing performance experiments are conducted on four types of rubber materials: FKM 1, FKM 2, FKM 3, and HNBR. The mechanical properties and macroscopic morphology of the rubber samples are analyzed using an optical microscope, a rubber tensile testing machine, and a Shore hardness tester. Additionally, the corrosion and aging mechanisms of the rubber materials are investigated to provide reference values and theoretical foundations for selecting appropriate rubber cylinders and sealing rings for downhole tools operating under high-temperature, high pressure, and acidic conditions.

## 2. Materials and Methods 

High temperature and high pressure H_2_S/CO_2_ gas reservoirs are widely distributed in the Sichuan Basin (Sichuan, China). For example, the formation pressure of Yuanba gas field is 140 MPa, and the average content of H_2_S and CO_2_ is 5.14% and 7.5% [21]. The H_2_S and CO_2_ contents of Dengying Formation gas reservoir in Anyue Gas field are 1.18% and 6.41% [22]. The H_2_S and CO_2_ contents in Puguang Gas field are as high as 15.16% and 8.64% [23]. The H_2_S content and CO_2_ content in Marine carbonate strata in northeast Sichuan range from 12.31% to 17.05% and from 7.89% to 10.53% [24,25]. The H_2_S content of Weiyuan Sinian gas reservoir is as high as 32.38%, and that of Wolong River Jialingjiang Formation is about 13.48% [26]. Therefore, downhole tools are affected by both by high temperature and high pressure, and also by strong corrosive media. Their corrosive environment and characteristics are more complex than those of conventional acid gas wells, which puts forward more stringent requirements for the design of corrosion test equipment and environment. Therefore, according to reservoir characteristics and rubber working environment of downhole tools, this paper designed corrosion rate measurement experiments suitable for different production stages in high temperature and high-pressure corrosion environments containing H_2_S and CO_2_, and carried out corrosion law research.

### 2.1. Corrosion of Experimental Rubber Material

The basic formulations of experimental fluororubbers 1, 2, 3 and HNBR rubber samples are shown in Table 1:

### 2.2. Corrode the Experimental Medium

According to the results of formation gas chromatography in Well X of the Sichuan Basin, the gas groups can be divided into 5%H_2_S, 20%CO_2_ and 75%CH_4_. The formation water environment is in accordance with the salinity analysis results provided on site. The total salinity was 76,090 mg/L, the water type was CaCl_2_, the PH value was 6.57, the cations contained were K^+^, Na^+^, Ca^2+^, Ba^2+^, Sr^2+^, and the anions mainly contained were Cl^−^. The specific ion content is shown in Table 2.

According to the above ion content, it can be calculated that the chemical reagents needed to prepare the above simulated formation aqueous solution are mainly: NaCl, KCl, MgCl_2_, SrCl_2_, CaCl_2_, BaCl_2_, in which the amount of NaCl is 20.1832 g/L, the amount of KCl is 17.9293 g/L, the amount of MgCl_2_ is 7.9231 g/L, and the amount of SrCl_2_ is 0.2370 g/L. The amount of CaCl_2_ was 30.5333 g/L, and the amount of BaCl_2_ was 1.3116 g/L.

### 2.3. Sample Processing and Size Requirements for the Experiment

According to the standard GB/T 531.1-2008 (China) [27], GB/T 528-2009 (China) [28], O-ring, hardness sample, and dumbbell tensile samples are needed for evaluating the corrosion resistance of rubber materials of downhole tools.

(1)O-ring sample

The dimensional accuracy of ϕd_1_ = 47.2 mm and ϕd_2_ = 3.6 mm in the O-ring meets the standard tolerance requirements of the rubber O-ring. Test 3 parallel samples in each group to observe the sealing status of the samples after the test and whether the PH paper changes, so as to evaluate the sealing ability and corrosion resistance of the rubber seal material under pressure, as shown in Figure 1.

(2)Hardness sample

Hardness samples were required to be plate samples with a thickness of ≥6 mm and length and width ≥ 30 mm, the specific size can be determined according to the experimental requirements. Each group of experiments had 3 parallel samples, and the samples without experiment are used to test the rubber Shore A hardness before the experiment. After the experiment, the average value of Shore A hardness was obtained by taking photos to observe and test. This study mainly tested the effect of corrosion on the Shore hardness of rubber material of downhole tools.

(3)Dumbbell-type sample

Dumbbell-type tensile specimens were tested with 3 parallel specimens in each group to test the tensile strength and elongation at break after the test, mainly to test the influence of corrosion on the tensile properties of rubber. Refer to GB/T 528-2009 (China), as shown in Figure 2 [28].

### 2.4. Instruments for Corrosion Experiments

Currently, the majority of indoor corrosion experiments are conducted in agitated high-temperature autoclaves, yielding experimental results that significantly deviate from field practices. Firstly, due to structural defects in the agitated tank body, the corrosive medium is unable to flow freely. Secondly, solid particles within the corrosive medium precipitate to the bottom of the vessel under gravity and fail to cleanse the sample surface. Additionally, as a result of its inherent viscosity, it becomes challenging to accurately determine the flow rate of the corrosive medium since stirring rate does not equate with fluid flow rate. To address these issues, a dynamic cyclic multiphase flow high-temperature autoclave capable of reaching maximum pressures and temperatures up to 150 MPa and 350 °C, respectively is employed for simulating actual temperature and pressure conditions at 90 MPa/80 °C and 90 MPa/160 °C as depicted in Figure 3. The high-temperature and high-pressure reactor was developed by Southwest Petroleum University (Model: GF-150-350-6HC) [29]. The equipment can simulate the corrosive environment of underground high-temperature and high-pressure fluid flow by heating equipment, blade stirring and gas injection. Compared with the conventional high-temperature and high-pressure reactor, it can simulate the working environment of underground tools more truly.

The rubber sealing device was used to measure the sealing corrosion of the rubber O-ring under pressure. The deformation characteristics of the O-ring after the experiment are observed with a stereo microscope. A Shore hardness tester was used to measure Shore hardness of the block and plate samples after corrosion test. The rubber tensile testing machine was used to test the tensile properties of dumbbell samples, and the tensile strength and elongation at break of dumbbell samples were calculated according to GB/T 528-2009 “Determination of tensile stress and strain properties of vulcanized rubber or thermoplastic rubber” (China) standard, so as to determine the performance of rubber.

(1)Calculation formula for permanent deformation of tensile break:

(1)$$S_{b}=\frac{100\left(L_{t}-L_{0}\right)}{L_{0}}$$ where S_b_ is the tensile permanent deformation percentage; L_t_ is the calibration distance of the specimen after it is placed for 3 min after fracture in mm; and L_0_ is the initial experiment length in mm.

(2)Tensile strength (MPa):

(2)$$Ts=\frac{F_{m}}{Wt}$$ where F_m_ is the record the maximum force in N; t is the test length part thickness in mm; and W is the width of the narrow part of the knife in mm.

(3)Elongation at break (%):

(3)$$E_{b}=\frac{L_{b}-L_{0}}{L_{0}}\times 100\%$$ where E_b_ is the elongation at break percentage; L_0_ is the initial test length, mm; and L_b_ is the test length at break in mm.

### 2.5. Corrosion Test Procedure

In accordance with the standard GB/T 531.1-2008) (China) [27], GB/T 528-2009 (China) [28] standards, processing rubber material corrosion resistance of O-rings, hardness samples and dumbbell tensile samples required for experimental evaluation. It is equipped with simulated formation water with water type of CaCl_2_ and salinity of 7.6 × 10^4^ mg/L. a high-temperature autoclave was used to test the liquid phase corrosion resistance of four kinds of rubber materials, FKM 1, FKM 2, FKM 3, and HNBR. The experiment period was 7 days, and the influence of temperature changes of 80 °C and 160 °C on the corrosion resistance of rubber materials was verified.

Before the experiment, a vernier caliper was used to measure the width × thickness of the distance section of the dumbbell sample for the subsequent drawing of the tensile stress–strain curve. In the experimental preparation stage, O-rings of four materials were installed on the sealing device as shown in Figure 4; for each test material, 3 parallel samples were taken to ensure the accuracy of the experiment. Dumbbell samples, O-ring samples placed in the sealing device, test block samples and plate samples were put into the high-temperature autoclave, and the pre-equipped simulated formation aqueous solution was poured into the high-temperature autoclave to ensure that the samples were submerged in the solution and there was still some space between the solution and the top of the kettle. Finally, cover the kettle.

During the pressurization phase of the experiment, an N_2_ pressure test was carried out to check the tightness of the reactor body and deaerate it. After deoxygenation, the temperature is raised to the required experimental temperature (80 °C, 160 °C), 20% CO_2_ and 5% H_2_S gas are injected, and N_2_ is pressurized to the required experimental pressure of 90 MPa. During the experiment, the temperature and pressure values of the high-temperature autoclave were monitored and recorded by computer to ensure the stability of temperature and pressure.

After 7 days of cooling and pressure relief, the sample was taken out and dried in cold air and photographed. The appearance and morphology characteristics of the rubber after corrosion test were observed and recorded. Then, the stereomicroscope was used to observe whether the O-ring bubble swelling occurred after the experiment. Shore hardness tester was used to measure the Shore A hardness of block and plate samples after a corrosion test. The rubber tensile testing machine uses the tensile speed of 500 mm/min to break the dumbbell rubber sample, record the breaking point, measure the maximum load value during breaking, and measure the standard distance of the parallel segment of dumbbell rubber after breaking and the stress–strain curve during the breaking process, so as to calculate the corrosion resistance and corrosion resistance parameters according to the standard GB/T 528-2009 (China).

## 3. Results and Discussion

The O-ring was used to determine the rubber sealing performance and observe the swelling characteristics of the specimen surface. The rubber test block was used to determine its hardness by the Shore hardness tester. The dumbbell sample was used for the tensile property test to determine its tensile strength, permanent deformation at break and elongation at break.

### 3.1. Hardness Properties of Different Rubber Materials

Fluororubber 1 appears to have large local bubbling at 80 °C, and dense small local bubbling at 160 °C. Fluororubber 2 was found to have obvious bubbling swelling under acidic pressure completion condition. Fluororubber 3 can be seen locally at a certain size of bubbling at a temperature of 80 °C, and no obvious bubbling swelling at a temperature of 160 °C. When the temperature of HNBR rubber is 80 °C, there is some small local bubbling, but no obvious bubbling or swelling phenomenon is seen at 160 °C, as shown in Figure 5.

Hardness characterizes the ability of rubber material to resist external deformation, the increase of rubber hardness will improve the strength and wear resistance of rubber products, have higher mechanical properties, and the resistance to oil, solvents and other chemical agents will be higher [30]. The hardness changes of various rubber materials at different temperatures are shown in Figure 6. The hardness of fluororubber 1 is most affected by temperature. The hardness decreases by 12.68% when the temperature is 80 °C and 17.48% when the temperature is 160 °C. The hardness of fluororubber 2 is not affected by temperature, and the hardness decreases within 1%; The hardness of fluororubber 3 decreases by 10.17% at 80 °C and 3.47% at 160 °C. The hardness of HNBR decreases by 2.41% when the temperature is 80 °C and 0.37% when the temperature is 160 °C. It can be seen that the hardness of HNBR and fluororubber 3 decreases at 160 °C, but the deformation resistance of fluororubber 3 is still poor. Fluororubber 1 has very poor resistance to deformation under acidic conditions, the aging resistance of fluororubber 2 is not affected by temperature, and the aging resistance of HNBR is better at high temperatures.

### 3.2. Tensile Properties of Different Rubber Materials

When the temperature of fluororubber 1 was 80 °C, small bubbles appeared locally, and when the temperature rose to 160 °C, the dumbbell sample was densely covered with small bubbles. Fluororubber 2 and HNBR rubber did not show obvious swelling phenomenon at the two temperatures. The local bubbling phenomenon occurred at the temperature of 80 °C, and the dumbbell sample of fluororubber 3 did not show obvious swelling phenomenon at the temperature of 160 °C, as shown in Figure 7.

In order to effectively analyze the performance changes of rubber materials before and after corrosion, the dumbbell rubber sample after corrosion test was pulled off by a special rubber tensile testing machine, and the rubber tensile stress–strain curve as shown in Figure 8 was drawn. According to the stress–strain curve, the average tensile strength and average tensile elongation at break of the four rubber materials could be calculated.

Elongation at break characterizes the crosslinking of the molecular structure of rubber materials, thus mapping the elongation properties of rubber materials [31]. Under simulated conditions, the elongation properties of the four kinds of rubber materials decreased significantly. With the increase of temperature, the elongation at break of fluororubber 1 and fluororubber 2 was positively correlated with the temperature. The elongation at break of fluororubber 1 and fluororubber 2 decreased by 1.34% and 26.49%, respectively at 160 °C compared with 80 °C. The elongation at break of HNBR and fluororubber 3 is inversely related to temperature, and the elongation at break of HNBR and fluororubber 3 at 80 °C is 21.22% and 4.14% lower than that at 160 °C, respectively. It can be seen that the molecular crosslinking structure of fluororubber 2 is greatly affected by temperature under the action of high temperature (160 °C), and the elongation property of fluororubber 1 is greatly reduced. The elongation of HNBR rubber at high temperature is better than that at 80 °C. The elongation properties of fluororubber 1 and fluororubber 3 are less affected by temperature, but the aging resistance is poor under this simulated working condition, as shown in Figure 9.

Tensile strength reflects the mechanical properties of rubber materials, and the higher the tensile strength, the greater the intermolecular cohesion of rubber materials [32]. The tensile strength of fluororubber 1 and fluororubber 2 decreased by 15.65% and 6.04%, respectively at 160 °C compared with 80 °C. The tensile strength of HNBR and fluororubber 3 decreased by 18.93% and 42.75%, respectively at 80 °C compared with 160 °C. The mechanical properties of fluororubber 3 are most affected by temperature, followed by HNBR, fluororubber 1 and fluororubber 2, as shown in Figure 10.

### 3.3. Sealing Performance of Different Rubber O-Rings

The sealing device shown in Figure 4 was used to conduct the rubber O-ring sealing corrosion test. After 7 days of corrosion, it was found that all the sealing devices were filled with water, and different rubber materials showed different degrees of deformation under different temperatures.

As shown in Figure 11, Fluororubber 2 and HNBR were slightly deformed at 80 °C and 160 °C, and no obvious swelling and bubbling phenomenon was observed. At 80 °C, the outer side of fluorine rubber 1 appeared barb and local swelling cracking phenomenon, and at 160 °C, the rubber bubbled and burst, and a large size crack appeared. Fluororubber 3 bubbled and swelled on both sides at 80 °C, small cracks appeared on the inside, and a large number of barbs appeared on the outside of the rubber at 160 °C. It can be preliminarily concluded that at 80 °C and 160 °C, fluororubber 2 and HNBR have better temperature resistance and pressure resistance than fluororubber 1 and fluororubber 3.

### 3.4. Experimental Results of Corrosion Resistance of Rubber Material

According to the experiment, the presence of H_2_S and CO_2_ in high-temperature and high-pressure environments will lead to the decay of rubber material properties, and the influence degree of rubber aging properties at different temperatures is inconsistent. Under the influence of high temperature, high pressure, and acidic medium, the molecular chain breaks, the cross-linked network structure of rubber molecules is damaged, and the mechanical properties of rubber materials are reduced. The O-rings of fluororubber 1 and fluororubber 3 have cracks and barbs, and their anti-aging properties are poor. Fluororubber 2 and HNBR rubber have good corrosion resistance. In the same corrosive medium, the effect of high temperature (160 °C) leads to further crosslinking of rubber material aging; As the degree of crosslinking increases, the cross-linking network is dense, and the dense cross-linking network makes the rubber molecules more compact [33,34], so the tensile properties and hardness of HNBR and fluororubber 3 at 160 °C are higher than those at 80 °C.

According to the corrosion resistance test of rubber material, as shown in Figure 12; The bubbling rubber test block has a large hardness reduction, such as fluorine rubber 1; Fluororubber 2 is not affected by temperature, and the decrease is less than 1%; The hardness of HNBR is less affected by temperature. Fluorine rubber 3 at 80 °C hardness reduction of up to 10%. According to the corrosion resistance test of rubber dumbbell samples, the crosslinking performance of four kinds of rubber materials is greatly affected under simulated working conditions, and the elongation at break is reduced by nearly or more than half compared with non-corroded rubber. Moreover, the elongation at break of HNBR and fluororubber 2 is significantly affected by temperature, and the elongation at break of fluororubber 2 further decreases with the increase of temperature. The cross-linked structure of HNBR is more compact under high temperatures, so the elongation at break increases. The tensile strength of fluororubber 1 decreased the most under simulated acid condition. Fluororubber 2 is not affected by temperature, but the tensile strength decreases by nearly half under the acid simulation condition. The tensile strength of fluororubber 3 decreases greatly at 80 °C and less at 160 °C; The decrease of tensile strength of HNBR is small, and the decrease of tensile strength at 160 °C is lower than that at 80 °C. According to the rubber O-ring corrosion resistance test, the O-ring deformation of fluorine rubber 1 material is extremely serious, and the rubber has appeared large-size bubbling and cracking phenomenon; Fluororubber 3 can be seen bubbling and bursting at the temperature of 80 °C, and local barbs at the temperature of 160 °C. No obvious deformation was found in fluororubber 2 and HNBR.

To summarize, in the corrosive environment with high H_2_S and CO_2_ content at high temperature and pressure, fluororubber 1 and fluororubber 3 have poor aging resistance. Although fluororubber 2 has high deformation resistance, its tensile property is very poor under simulated acid conditions, so HNBR’s comprehensive performance is the best among the four rubber materials.

## 4. Conclusions

(1) After conducting comprehensive testing on the corrosion resistance (including hardness, tensile strength, elongation at break, and seal of O-type rubber ring) of fluororubber 1, fluororubber 2, fluororubber 3, and HNBR under simulated conditions of high temperature and high pressure with H_2_S and CO_2_ gases, it is evident that both fluororubber 1 and fluororubber 3 exhibit significant deformation. On the other hand, HNBR and fluororubber 2 demonstrate good deformation resistance; however, the tensile strength of fluororubber 2 decreases significantly at both temperatures. The cross-linking reaction between HNBR and fluororubber 3 occurs due to temperature influence resulting in better corrosion resistance at a temperature of 160 °C compared to that at an 80 °C. Furthermore, all four types of rubber materials experience a substantial decrease in elongation at break. 

(2) The performance characteristics of rubber materials are influenced by their molecular structure. In corrosive environments with elevated levels of H_2_S and CO_2_ under high-temperature and high-pressure conditions, physical as well as chemical reactions take place between the rubber material itself and the corrosive medium leading to changes in its molecular structure. Consequently, this aging process causes a reduction in material properties. 

(3) Through evaluating the corrosion resistance capabilities of four different types of rubber materials under acidic pressing conditions at various temperatures, it can be observed that the corrosion resistance ability of fluororubber 3 is greatly affected by temperature fluctuations while the other three types show less sensitivity towards such variations. The overall performance quality ranks highest for HNBR, universally followed by fluororubber 2, and then fluororubber 3. Fluororubber 1 exhibits comparatively inferior performance. This provides valuable theoretical insights for optimizing rubber material selection under acidic conditions.

## Figures and Tables

**Figure 1 materials-17-00863-f001:**
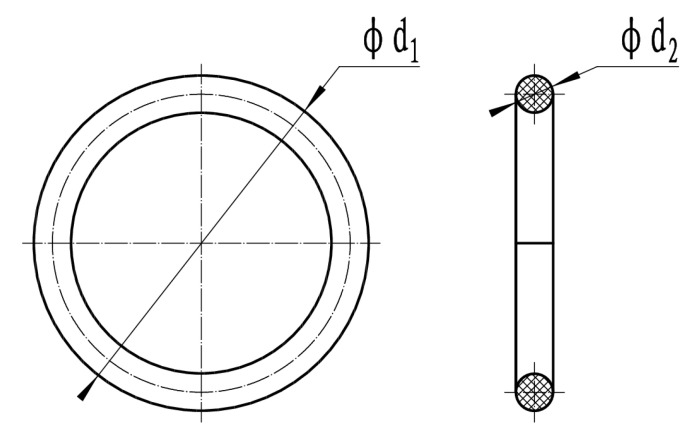
Size requirements of rubber O-ring sample.

**Figure 2 materials-17-00863-f002:**
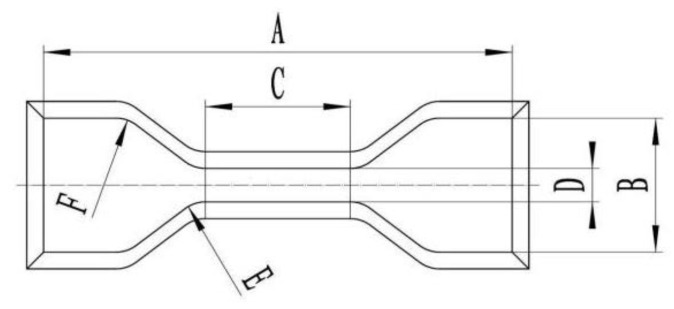
Size requirements of dumbbell sample made of rubber material. Note: The dimensions of A~F are shown in Table 2. 1—Tool holder head fixed to the supporting machine. 2—Need grinding; 3—Need polishing.

**Figure 3 materials-17-00863-f003:**
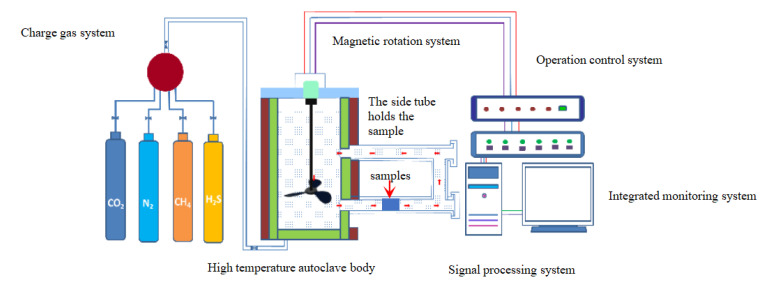
Schematic diagram of dynamic cyclic corrosion experimental device for high-temperature and high-pressure multiphase flow [29].

**Figure 4 materials-17-00863-f004:**
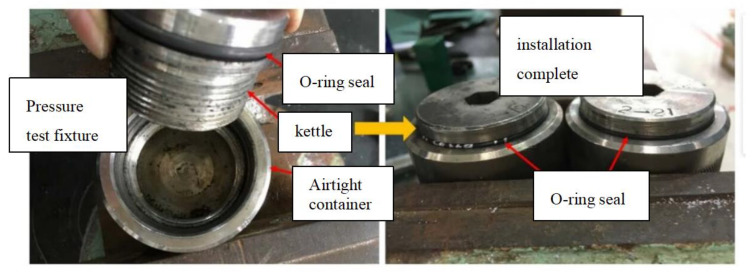
Rubber O-ring sealing device.

**Figure 5 materials-17-00863-f005:**
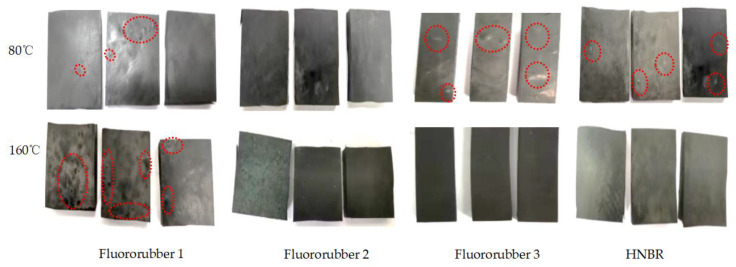
Macroscopic morphologies of test blocks and plates after corrosion for 7 days with different rubber materials at different temperatures.

**Figure 6 materials-17-00863-f006:**
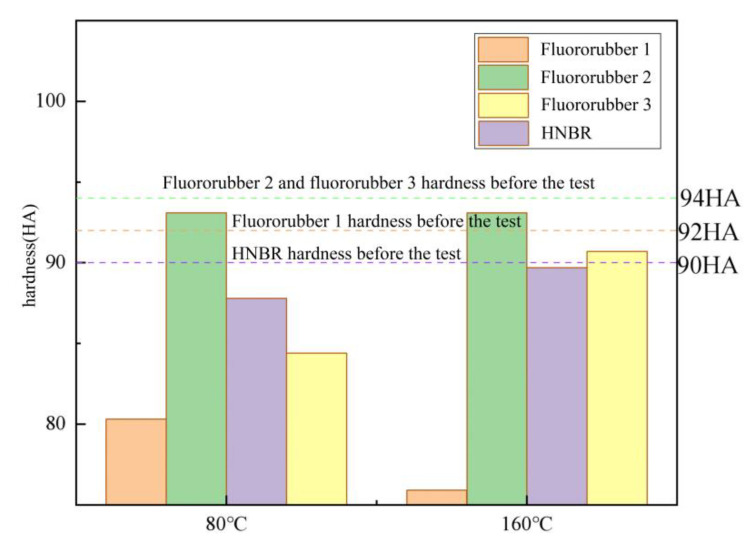
Hardness and properties of different rubber materials corroded for 7 days at different temperatures.

**Figure 7 materials-17-00863-f007:**
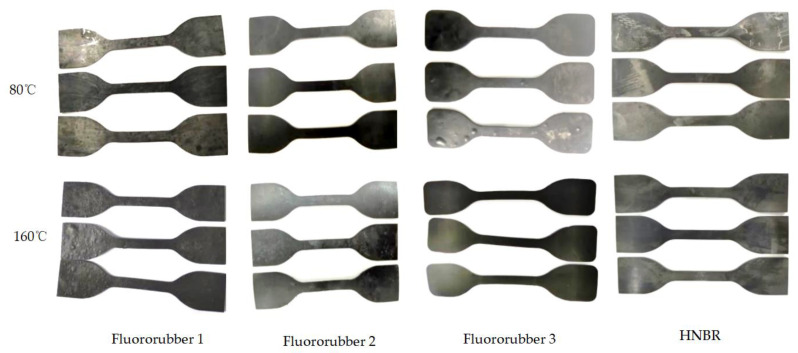
Macroscopic morphology of dumbbell samples corroded for 7 days under different rubber materials and different temperatures.

**Figure 8 materials-17-00863-f008:**
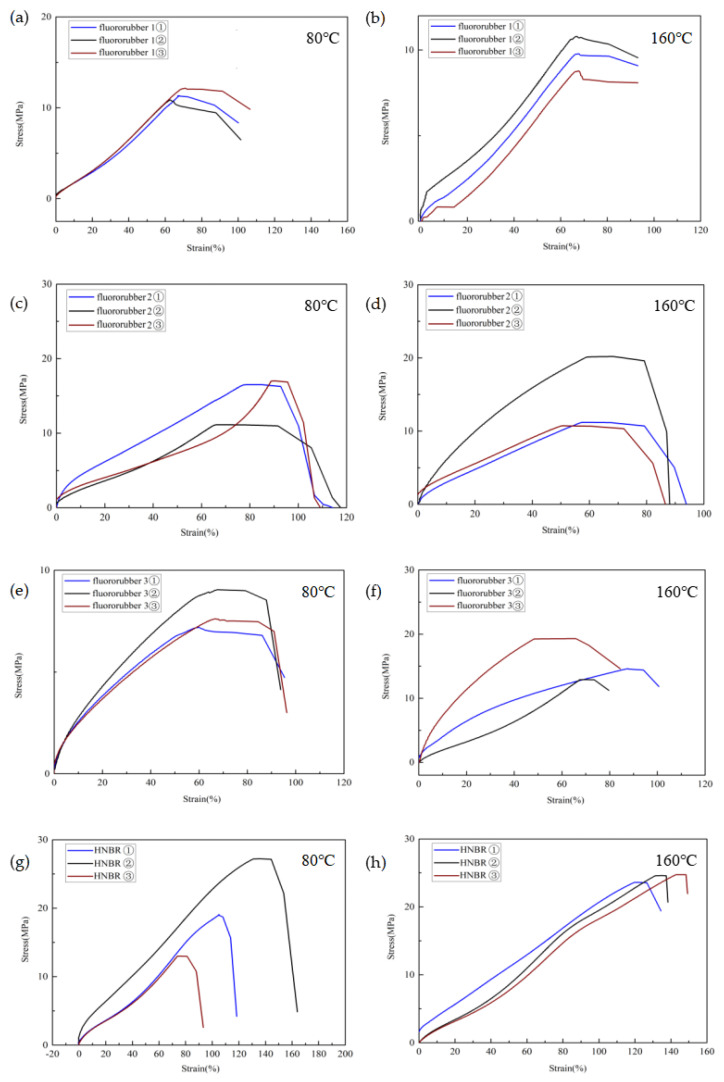
Stress–strain curves of four kinds of rubber corroded at different temperatures for 7 days. (**a**) Stress-strain curve of fluororubber 1 at 80 °C; (**b**) Stress-strain curve of fluororubber 1 at 160 °C; (**c**) Stress-strain curve of fluororubber 2 at 80 °C; (**d**) Stress-strain curve of fluororubber 2 at 160 °C; (**e**) Stress-strain curve of fluororubber 3 at 80 °C; (**f**) Stress-strain curve of fluororubber 3 at 160 °C; (**g**) Stress-strain curve of HNBR at 80 °C; (**h**) Stress-strain curve of HNBR at 160 °C.

**Figure 9 materials-17-00863-f009:**
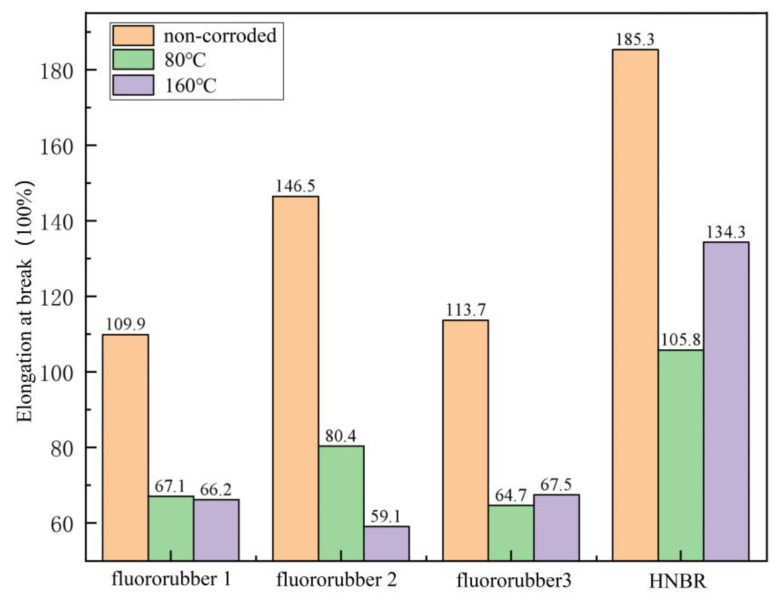
Elongation at break after 7 days of corrosion of rubber material at different temperatures.

**Figure 10 materials-17-00863-f010:**
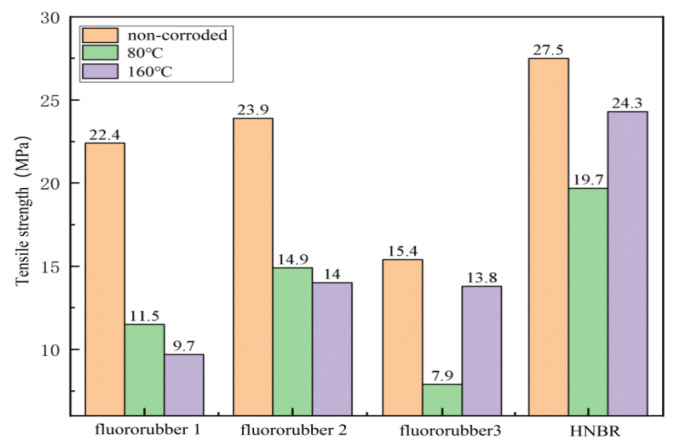
Tensile strength diagram of rubber material corroded for 7 days at different temperatures.

**Figure 11 materials-17-00863-f011:**
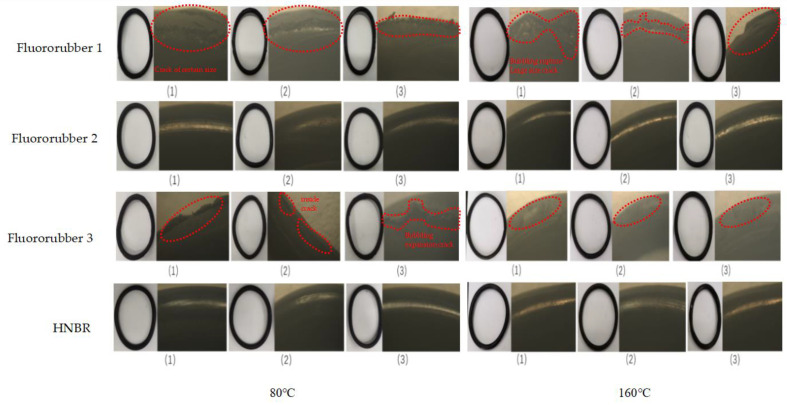
Macro topography of rubber O-ring.

**Figure 12 materials-17-00863-f012:**
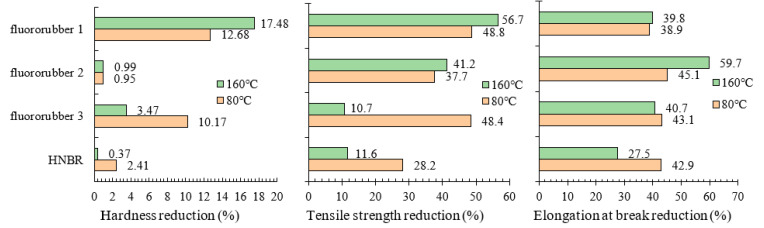
Performance decline of four kinds of rubber materials after corrosion test.

**Table 1 materials-17-00863-t001:** Rubber name and basic formula table.

Name	Basic Formula (Mass Fraction)	Supplier
Fluororubber 1	Tetrapropyl fluororubber is 100, carbon black is 40, Bisphenol AF is 5, BPP is 2.	Shanghai Truthful sealing Technology Co., Ltd. (Shanghai, China)
Fluororubber 2	Fluorine rubber is 100, carbon black is 40, Bisphenol AF is 2, BPP is 2.	Shanghai Truthful sealing Technology Co., Ltd. (Shanghai, China)
Fluororubber 3	Perflurane rubber is 7, tetrapropyl fluorine rubber is 48, fluorine silicone rubber is 48, BIBP is 2, Cycloxy promoter is 2.	Hebei Cheneng Technology Co., Ltd. (Hebei, China)
HNBR	HNBR is 100, carbon black is 60, Stearic acid is 2, Zinc oxide is 2.	Shanghai Truthful sealing Technology Co., Ltd. (Shanghai, China)

**Table 2 materials-17-00863-t002:** Analysis data of formation water quality under working conditions of rubber seals of downhole tools.

Serial Number	Ion Species	Ion Content, mg/L	Serial Number	Ion Species	Ion Content, mg/L
1	K^+^ + Na^+^	14,940	4	Ba^2+^	865
2	Ca^2+^	9227	5	Sr^2+^	131
3	Mg^2+^	2023	6	Cl^−^	46,728

## Data Availability

Data are contained within the article.

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
