# Peer review of "Characterization and Analysis of Corrosion Resistance of Rubber Materials for Downhole Tools in a High-Stress Environment with Coupled H2S-CO2"

_materials, 2024, doi:10.3390/ma17040863_

Round 1
Reviewer 1 Report
Comments and Suggestions for Authors
Manuscript Materials-2740014 entitled "Characterization and Analysis of Corrosion Resistance of Rubber Materials for Downhole Tools in a High-Stress Environment with Coupled H2S-CO2".
In this manuscript, the authors studied rubber materials under high-temperature and high-pressure conditions containing H2S and CO2 to simulate a corrosive environment. They reported the properties obtained after aging for different rubbers. The authors show interesting results for industrial rubbers and rubbers for applications for downhole tools and ring seals. However, revisions would be needed to have the manuscript may be considered for publication in Materials.
My comments:
1. In the Abstract, define HNBR.
2. Authors are recommended to add 2 or 3 more Keywords.
3. In the introduction section, the authors are recommended to briefly explain the different types of industrial rubbers (NR, BR, SBR, NBR, HNBR, FKN, FEPM), their applications and properties in general. Since these share similar applications and properties, which are mentioned in the text. To later specify the rubbers with which you work in the manuscript (FKN and HNBR).
Some references are recommended for the different types of rubber, their properties and structures:
a). Synthesis of Biobased Hydroxyl-Terminated Oligomers by Metathesis Degradation of Industrial Rubbers SBS and PB: Tailor-Made Unsaturated Diols and Polyols. Polymers. 2022, 14, 4973. https://doi.org/10.3390/polym142249733
b). Synthetic Polyisoprene Rubber as a Mimic of Natural Rubber: Recent Advances on Synthesis, Nanocomposites, and Applications. Polymers. 2023, 15, 4074. https://doi.org/10.3390/polym15204074
4. It is recommended that recent references in the introduction (2021-2023) be added. There are no 2023 references in the manuscript.
5. Page 1, line 41. The authors mention "Currently, Yang et al" but the reference is from 2012. This information needs to be updated.
6. Page 2, line 51. Define AFLAS.
7. Page 2, lines 87-88. The names and abbreviations of rubbers were already mentioned above; use the abbreviations.
8. Page 3. In section 2, it is recommended to add the main title of "Materials and Methods" and use the corresponding subtitles for the others (2.1, 2.2, 2.3...).
9. In this section 2, it is still necessary to indicate:
a). Reagents and materials used.
b). Equipment used (description, model, operating conditions).
Since it is unclear what was used in the research, Where did the rubber come from? Give more information.
What is the difference between fluororubber 1, 2 and 3? Why are 3 used? Since this part needs to be clarified. Mention in the text.
10. Page 4, section 2.2. The authors mention some standards used for characterization, but they are not international standards. Are there any international standards or their equivalent that have been used? such as ISO or ASTM. This is so that other countries can reproduce the analyses or compare results.
11. Are Figure 2 and Table 2 reported in the standard? Since they are universal parameters included in the standard, making a figure and table in the manuscript is not recommended. The authors can only mention that the measurements of the test pieces were according to the XXYY standard.
12. Page 6, line 194. Put Fig. 4 after it is mentioned.
13. Page 6. See the format for the equations in the journal and the numbering. Correction is recommended.
14. Page 7. In section 3, it is recommended to add the main title of "Results and Discussion" and use the corresponding subtitles for the others (3.1, 3.2, 3.3...).
15. Page 8, improve the quality of Fig. 5. Indicate which FKM 1, 2 and 3 correspond to.
16. Pages 8-9. Mention Fig. 7 in the text.
17. Page 9, Figure 7. What does rubber look like before aging? Some results do not compare to pure rubber or before aging.
18. Page12, line 328. Are the authors referring to Fig. 4 or Fig. 11?
Improve the quality of Fig. 11, indicating the type of rubber (FKM1, 2, 3). Some letters or text need to be visible in the figure.
19. All references must be in the same format in the References section. Review the journal format (reference list). Some references have letters in brackets [J] and [D]. Reviewing and correcting this section is recommended.
Current references (2022-2023) are recommended.
It is also recommended that references from this journal (Materials) be cited.
Author Response
My comments:
- In the Abstract, define HNBR.
Dear reviewer, it has been modified according to your requirements.
- Authors are recommended to add 2 or 3 more Keywords.
Dear reviewer, it has been modified according to your requirements.
- In the introduction section, the authors are recommended to briefly explain the different types of industrial rubbers (NR, BR, SBR, NBR, HNBR, FKN, FEPM), their applications and properties in general. Since these share similar applications and properties, which are mentioned in the text. To later specify the rubbers with which you work in the manuscript (FKN and HNBR).
Some references are recommended for the different types of rubber, their properties and structures:
a). Synthesis of Biobased Hydroxyl-Terminated Oligomers by Metathesis Degradation of Industrial Rubbers SBS and PB: Tailor-Made Unsaturated Diols and Polyols. Polymers. 2022, 14, 4973. https://doi.org/10.3390/polym142249733
b). Synthetic Polyisoprene Rubber as a Mimic of Natural Rubber: Recent Advances on Synthesis, Nanocomposites, and Applications. Polymers. 2023, 15, 4074. https://doi.org/10.3390/polym15204074
Dear reviewer, it has been modified according to your requirements.
- It is recommended that recent references in the introduction (2021-2023) be added. There are no 2023 references in the manuscript.
Dear reviewer, it has been modified according to your requirements.
- Page 1, line 41. The authors mention "Currently, Yang et al" but the reference is from 2012. This information needs to be updated.
Dear reviewer, where I expressed accurately, I have changed it to "recent years".
- Page 2, line 51. Define AFLAS.
Dear reviewer, it has been modified according to your requirements.
- Page 2, lines 87-88. The names and abbreviations of rubbers were already mentioned above; use the abbreviations.
Dear reviewer, it has been modified according to your requirements.
- Page 3. In section 2, it is recommended to add the main title of "Materials and Methods" and use the corresponding subtitles for the others (2.1, 2.2, 2.3...).
Dear reviewer, it has been modified according to your requirements.
- In this section 2, it is still necessary to indicate:
a). Reagents and materials used.
b). Equipment used (description, model, operating conditions).
Since it is unclear what was used in the research, Where did the rubber come from? Give more information.
What is the difference between fluororubber 1, 2 and 3? Why are 3 used? Since this part needs to be clarified. Mention in the text.
Dear reviewer, it has been modified according to your requirements.
- Page 4, section 2.2. The authors mention some standards used for characterization, but they are not international standards. Are there any international standards or their equivalent that have been used? such as ISO or ASTM. This is so that other countries can reproduce the analyses or compare results.
Dear reviewer, there is no similar international standard at present, and the quoted standard is compiled according to the actual needs of China's oilfield field.
- Are Figure 2 and Table 2 reported in the standard? Since they are universal parameters included in the standard, making a figure and table in the manuscript is not recommended. The authors can only mention that the measurements of the test pieces were according to the XXYY standard.
Dear reviewer, it has been modified according to your requirements.
- Page 6, line 194. Put Fig. 4 after it is mentioned.
Dear reviewer, according to your suggestion, I put Figure 4 at the end.
- Page 6. See the format for the equations in the journal and the numbering. Correction is recommended.
Dear reviewer, it has been modified according to your requirements.
- Page 7. In section 3, it is recommended to add the main title of "Results and Discussion" and use the corresponding subtitles for the others (3.1, 3.2, 3.3...).
Dear reviewer, it has been modified according to your requirements.
- Page 8, improve the quality of Fig. 5. Indicate which FKM 1, 2 and 3 correspond to.
Dear reviewer, it has been modified according to your requirements.
- Pages 8-9. Mention Fig. 7 in the text.
Dear reviewer, it has been modified according to your requirements.
- Page 9, Figure 7. What does rubber look like before aging? Some results do not compare to pure rubber or before aging.
Dear reviewer, in the process of this experiment, three parallel samples were prepared for each sample. It can be seen from the samples after the experiment that the changes in the macro morphology of the sample after the experiment. The original samples were processed according to the same standard, with the same appearance and outline size, so the rubber photos before aging were not included.
- Page12, line 328. Are the authors referring to Fig. 4 or Fig. 11?
Dear reviewer, this refers to Figure 4. For the convenience of distinction, I have marked the words "as shown in Figure 11" in the following paragraph.
Improve the quality of Fig. 11, indicating the type of rubber (FKM1, 2, 3). Some letters or text need to be visible in the figure.
Dear reviewer, it has been modified according to your requirements.
- All references must be in the same format in the References section. Review the journal format (reference list). Some references have letters in brackets [J] and [D]. Reviewing and correcting this section is recommended.
Dear reviewer, letter [J] stands for journal, letter [D] stands for dissertation.
Current references (2022-2023) are recommended.
It is also recommended that references from this journal (Materials) be cited.
Dear reviewer, it has been modified according to your requirements.

Reviewer 2 Report
Comments and Suggestions for Authors
1. Fig. 8 can be improved and comparable scale needs to be used. For reference, see the following article:
https://www.sciencedirect.com/science/article/abs/pii/S0142941820320857
2. Concerning failure of O rings shown in Fig. 11 (topography of O rings), the correct explanations in terms of failure / fracture / durability of rubber composites need to be given. Some explanations can be found in:
https://www.sciencedirect.com/science/article/abs/pii/S1350630723003126
Comments on the Quality of English LanguageThe English language can be improved.
Author Response
- Fig. 8 can be improved and comparable scale needs to be used. For reference, see the following article:
https://www.sciencedirect.com/science/article/abs/pii/S0142941820320857
Dear reviewer, it has been modified according to your requirements.
- Concerning failure of O rings shown in Fig. 11 (topography of O rings), the correct explanations in terms of failure / fracture / durability of rubber composites need to be given. Some explanations can be found in:
https://www.sciencedirect.com/science/article/abs/pii/S1350630723003126
Dear reviewer, it has been modified according to your requirements.

Reviewer 3 Report
Comments and Suggestions for Authors
The article describes the corrosion resistance of various O-ring rubbers under the conditions of gas extraction. The introduction to previous work is sufficient and helpful. Also the choice of experiments and tests is sound and justified.
However the description of samples and the test results are, in the present form unsufficient to the average reader (maybe not the case for very specialists).
The authors must provide a table specifying all materials under test, their base chemical composition, the manufacturer, the processing batch and the additives used therein (if avaiable) as well as the manufacturer's specification in terms of corrosion resistance and a valid link to the specific datat sheets. This must be made the basis of the entire discussion.
Moreover, HBNR is first explained in line 88 while occurring already earlier incl. abstract.
Naming fluororubber 1...3 without specifying is scientifically unsufficient.
In fig 3 the sample not visible.
Fig 5 and others are missing "1" ... "3"
Line 270: Specify the method of hardness testing and relate to classic properties.
Fig 8 and other: Are there standard deviations? Please use, at least for the x-axis, standard scaling in order to ease comparison.
Fig 11 missing description.
Overall discussion: Why are several corrosion / stress results higher at 80 °C than measured at 160 °C ??? What can be learned from the materials compositions in terms of corrosion resistance parameters and how do these findings comply with previous findings cited in the introduction? Hence, how advances this work the scientific state of the art?
Comments on the Quality of English LanguageAlmost fine.
Author Response
The article describes the corrosion resistance of various O-ring rubbers under the conditions of gas extraction. The introduction to previous work is sufficient and helpful. Also the choice of experiments and tests is sound and justified.
However the description of samples and the test results are, in the present form unsufficient to the average reader (maybe not the case for very specialists).
The authors must provide a table specifying all materials under test, their base chemical composition, the manufacturer, the processing batch and the additives used therein (if avaiable) as well as the manufacturer's specification in terms of corrosion resistance and a valid link to the specific datat sheets. This must be made the basis of the entire discussion.
Dear reviewer, it has been modified according to your requirements.
Moreover, HBNR is first explained in line 88 while occurring already earlier incl. abstract.
Dear reviewer, it has been modified according to your requirements.
Naming fluororubber 1...3 without specifying is scientifically unsufficient.
Dear reviewers, because the three total rubber belongs to a kind of fluorine rubber, the rubber is named fluorine rubber 1,2,3.
In fig 3 the sample not visible.
Dear reviewer, it has been modified according to your requirements.
Fig 5 and others are missing "1" ... "3"
Dear reviewer, it has been modified according to your requirements.
Line 270: Specify the method of hardness testing and relate to classic properties.
Dear reviewer, it has been modified according to your requirements.
Fig 8 and other: Are there standard deviations? Please use, at least for the x-axis, standard scaling in order to ease comparison.
Dear reviewer, Figure 8 is the stress-strain curve of 4 kinds of rubber corroded at different temperatures for 7 days. The X-axis represents the percentage of the dependent variable, which is the standard scale.
Fig 11 missing description.
Dear reviewer, it has been modified according to your requirements.
Overall discussion: Why are several corrosion / stress results higher at 80 °C than measured at 160 °C ??? What can be learned from the materials compositions in terms of corrosion resistance parameters and how do these findings comply with previous findings cited in the introduction? Hence, how advances this work the scientific state of the art?
Dear reviewer
The performance characteristics of rubber materials are influenced by their molecular structure. In corrosive environments with elevated levels of H2S and CO2 under high temperature and pressure conditions, physical as well as chemical reactions take place between the rubber material itself and the corrosive medium leading to changes in its molecular structure. Consequently, this aging process causes a reduction in material properties.
Through evaluating the corrosion resistance capabilities of four different types of rubber materials under acidic pressing conditions at various temperatures, it can be observed that the corrosion resistance ability of fluororubber 3 is greatly affected by temperature fluctuations while the other three types show less sensitivity towards such variations.The overall performance quality ranks highest for HNBR universally followed by fluororubber 2 and then fluororubber 3. Fluororubber 1 exhibits comparatively inferior performance.This provides valuable theoretical insights for optimizing rubber material selection under acidic conditions.

Round 2
Reviewer 1 Report
Comments and Suggestions for Authors
Manuscript ID Materials-2740014 entitled "Characterization and Analysis of Corrosion Resistance of Rubber Materials for Downhole Tools in a High-Stress Environment with Coupled H2S-CO2". The authors have addressed most of the comments. However, minor revisions would be needed to have the manuscript may be considered for publication in Materials.
My comments:
Some comments need to be addressed from the first revision, which are not reflected in the new version (v2):
1. In the introduction section. The authors are recommended to briefly explain the different types of rubbers (NR, BR, SBR, NBR, HNBR, FKN, FEPM), their applications and properties in general. Since these share similar applications and properties, which are mentioned in the text. To later specify the rubbers with which you work in the manuscript (FKN and HNBR). Some references are recommended for the different types of rubber, their properties and structures: Polymers 2022, 14, 4973. https://doi.org/10.3390/polym142249733. Polymers 2023, 15, 4074. https://doi.org/10.3390/polym15204074 and other references.
In the introduction section, lines 33-41. Putting more references in the text, not just one [1], is recommended since several types of rubber and different applications are addressed.
2. Materials and Methods.
In this section 2, it is still necessary to indicate:
a). Reagents and materials used.
b). Equipment used (description, model, operating conditions).
c) Since it is unclear what was used in the research, Where did the rubber come from? Give more information. Supplier.
Author Response
Manuscript ID Materials-2740014 entitled "Characterization and Analysis of Corrosion Resistance of Rubber Materials for Downhole Tools in a High-Stress Environment with Coupled H2S-CO2". The authors have addressed most of the comments. However, minor revisions would be needed to have the manuscript may be considered for publication in Materials.
My comments:
Some comments need to be addressed from the first revision, which are not reflected in the new version (v2):
- In the introduction section. The authors are recommended to briefly explain the different types of rubbers (NR, BR, SBR, NBR, HNBR, FKN, FEPM), their applications and properties in general. Since these share similar applications and properties, which are mentioned in the text. To later specify the rubbers with which you work in the manuscript (FKN and HNBR). Some references are recommended for the different types of rubber, their properties and structures: Polymers 2022, 14, 4973.
https://doi.org/10.3390/polym142249733. Polymers 2023, 15, 4074. https://doi.org/10.3390/polym15204074 and other references.
In the introduction section, lines 33-41. Putting more references in the text, not just one [1], is recommended since several types of rubber and different applications are addressed.
Dear reviewer, it has been modified according to your requirements.
- Materials and Methods.
In this section 2, it is still necessary to indicate:
a). Reagents and materials used.
b). Equipment used (description, model, operating conditions).
- c) Since it is unclear what was used in the research, Where did the rubber come from? Give more information. Supplier.
- a) Dear reviewer, the experimental liquid used is shown in 2.2
- b) Dear reviewer, it has been modified according to your requirements.
- c)Dear reviewer, it has been modified according to your requirements.

Reviewer 3 Report
Comments and Suggestions for Authors
The authors have now made very helpful changes. It is now realized that the authors prepared the sample materials by themselves. Hence, the starting materials (supplier companies and article numbers) should be specified in a way that the reader is able to re-do these experiments. Statements like "vulcanization agent" are insufficient. Please, also describe the methods and parameters of polymer fabrication.
All other points are fine now.
Author Response
The authors have now made very helpful changes. It is now realized that the authors prepared the sample materials by themselves. Hence, the starting materials (supplier companies and article numbers) should be specified in a way that the reader is able to re-do these experiments. Statements like "vulcanization agent" are insufficient. Please, also describe the methods and parameters of polymer fabrication.
All other points are fine now.
Dear reviewer
According to your requirements, the information of vulcanizing agent and supplier has been clarified, but we are sorry that we cannot provide the manufacturing process of rubber. The supplier's reason is that the manufacturing process of the whole rubber process is a technical secret and cannot be provided. I'm really sorry.
